# Understanding the Wellbeing of the Oldest-Old in China: A Study of Socio-Economic and Geographical Variations Based on CLHLS Data

**DOI:** 10.3390/ijerph16040601

**Published:** 2019-02-19

**Authors:** Lijuan Gu, Yang Cheng, David R. Phillips, Mark Rosenberg

**Affiliations:** 1Key Laboratory of Land Surface Pattern and Simulation, Institute of Geographic Sciences and Natural Resources Research, China Academy of Sciences, Beijing 100101, China; gulj@igsnrr.ac.cn; 2Faculty of Geographical Science, Beijing Normal University, Beijing 100875, China; 3Department of Sociology and Social Policy, Lingnan University, Hong Kong 999077, China; phillips@LN.edu.hk; 4Department of Geography and Planning, Queen’s University, Kingston, ON K7L3N6, Canada; mark.rosenberg@queensu.ca

**Keywords:** oldest-old, health, quality of life, socioeconomic factors, geography, China

## Abstract

Empirical studies of the socio-economic determinants of the wellbeing of the oldest-old in China including the role of geography and spatial factors are rare. This paper applies binary logistic regression analysis to data on the oldest-old aged 80 years old and higher from the 2011 Chinese Longitudinal Healthy Longevity Study (CLHLS). Socioeconomic determinants of the self-reported quality of life (QoL) and self-reported health (SRH) of the oldest-old population are explored, with special attention paid to the role of residence and region. The results indicate that, after controlling for individual demographic and health behavior variables, both economic status and social welfare have a significant effect on self-reported QoL and SRH. There are also significant differences in self-reported QoL among cities, towns and rural areas, with the oldest-old respondents living in Central rural, Western town and Western rural areas being significantly less likely to report good QoL, compared to the oldest-old living in Eastern cities. Significant differences in SRH exist among Eastern China, Western China and Northeastern China, with the oldest-old from Western towns being significantly less likely to report good health, and the oldest-old from Northeastern cities being significantly more likely to report good health than those from Eastern cities. The results of this study indicate that socioeconomic factors that explain self-reported QoL and SRH of the older population are in general factors that explain the self-reported QoL and SRH of the oldest-old cohorts. The interaction effect of residence and region matters more than each of the individual factors, in providing us with more detailed information on the role of geography in explaining QoL and health of the oldest-old. At a time when the oldest-old cohorts in China are at the beginning of their projected growth, these findings are vital for providing policy makers with more information on the urgency of making more geographically targeted policy to improve more effectively the self-reported QoL and SRH of the oldest-old population.

## 1. Introduction

China has the largest absolute number of older people, and China is likely to be a leading country in proportions of the oldest-old persons in coming decades [1,2]. Due in large part to effects of the One-Child policy implemented from 1978 to 2015, China has experienced one of the most rapid processes of population aging. In 2000, China’s population over 60 years old reached 126 million, of whom the population aged 65 and over reached 86 million, accounting for 10% and just over 7% of the total population respectively [3]. The older population aged 65 and over reached 118.83 million, which accounted for 8.9% of the total population in 2010 [4]. According to a UN forecast, the population aged 65 and over in China will account for 17.2 % in 2030 and 27.6 % in 2050 [5]. Most importantly, persons aged 80+ (the oldest-old here) were 19 million in 2009, but are projected to increase considerably and reach 29 million in 2020 [6].

Increasing life expectancy has also played a part in rapid population aging in China. Life expectancy has increased significantly over the last three decades, with the national average rising from 69 in 1980 to 74.8 in 2010. An increase in the prevalence of chronic diseases has paralleled the increase in life expectancy of the older population. The National Health Service Survey in China (2013) reported that 71.8% of older people experienced chronic diseases. Especially among the oldest-old, the prevalence of instrumental activities of daily living (IADLs) problems and severe morbidity both increased in recent years [7]. The costs of lifespan extension are the increase in the older population surviving with health problems and their disabilities of physical and cognitive functioning according to a longitudinal study from 1998 to 2008 [8].

Many studies on health of older populations reported in the international literature focus on the environmental and social determinants of health. Age is often seen as the main factor in morbidity and impairment. However, factors such as education, health care, income and urbanization also relate to the prevalence of disability [7,9]. According to the National Population Health Survey of England, socio-economic indicators, particularly income support, were significantly associated with the odds of poor health outcomes. Social resources (such as marital status and social support) had the greatest effect on the indicator of psychological health and also contributed significantly to variation in SRH [10]. Factors such as advanced age, lower health status, greater number of chronic conditions, lower education, not being married and smoking are also associated with the probability of institutionalization and the determinants of health of people living in the community [11]. The health of older populations is apparently not only affected by individual factors, but political economy factors such as disparities in economic development and allocation of health care resources, create health inequities among the older population [12,13,14].

Geographers are particularly interested in understanding the regional differences in health status of the older populations, the impact of place on older people’s health and how effectively health services react to these differences and impacts. For example, a study on health status and health care use among the older Aboriginal population in Canada found that the place of residence affects their health status and utilization of health care services [15]. A study in Portugal concluded that the distribution of health service resources was an important way to understand the inequities of access to health and health care [16]. In terms of the impact of place, individuals living in materially deprived areas were more likely to be affected by health conditions resulting in restrictions of activity. Prevalence of psychological distress was higher in areas with greater social deprivation in a U.K. study [17]. Wilson et al. compared the effects of neighbourhood socioeconomic status on health status, health behaviours and health care use in Glasgow, Scotland and Hamilton, Canada [15]. The study suggests that country context may be important to explain the distribution of health status and health behaviours among socially contrasting neighbourhoods, and that neighbourhood variations in health may be context specific. 

In this respect, it is important to recognize that China is highly spatially differentiated in terms of its physical and cultural landscapes as well as economic development. There are also considerable regional differences in population ageing among provinces, as well as among rural and urban areas in China. In 2010, the ageing population ratio (APR, proportion of population aged 65 and over) ranged from 5.1 to 11.6 % among the 34 provincial divisions [2,18]. Moreover, in terms of rural and urban differences, the population aged 65 and over was 52.3 million in urban areas and 66.6 million in rural areas in 2010 [4]. The population ageing rate is generally higher and the process more rapid in rural areas than in urban areas in most of the provinces, often reflecting labour force migration from rural to urban areas, even though the fertility rate is higher in rural areas than in urban areas. The prevalence rate of chronic diseases is higher among the urban older population than the rural older population. Urban citizens generally also enjoy a range of social, economic and cultural benefits that the rural citizens do not due to the dual structure of the urban-rural system in China [19]. The change in population structure and health inequality among the older population will undoubtedly bring many challenges for the social welfare, health care and social care systems for the older population in the future, especially in the rural areas.

There have been few national scale studies of geographical variation in the health of the older population in China. Evandrou et al. used multi-level modeling to analyze data from the Chinese Health and Retirement Longitudinal Study (CHARLS) in order to investigate the individual, household and provincial characteristics associated with poor health among older people [20]. The results show that older Chinese women, rural residents, those with an education level lower than high school, without individual income sources, who are ex-smokers, and those from poor economic status households are more likely to report disability and poor self-rated health. Differentials in the health outcomes remain substantial between provinces even after controlling for a number of individual and household characteristics [5,21]. Feng et al. examined the geographical variations of SRH of the older population and how individual, family, and institutional factors affect their SRH at both individual and province levels based on the 2008 Chinese Longitudinal Healthy Longevity Survey (CLHLS) [22]. Results showed that uneven economic development, varied social security provision as well as different family support at regional levels all contributed to the geographical variations in health outcomes of the older population [22]. Recently, Feng et al. have explored regional and social differences in social exclusion of older persons in China based on the 2014 China Longitudinal Aging Social Survey (CLASS) data and the relations with health and well-being [12]. The results showed that older people with lower educational attainment, in the lowest quintile of personal income and in poor health were the most likely to be excluded. Significant unexplained variation for various dimensions of social exclusion was also found at the province level.

Studies focusing on the oldest-old population of China and social determinants of their health are even more limited. Wang et al. studied the spatial patterns and changes of the older population, the oldest-old and centenarians in China in 2000 and 2010 [6]. The results showed that proportions of the older population and the oldest-old are higher in regions with higher socio-economic development and that have a favorable climate. The distribution of centenarians is less influenced by economic factors. Lifestyle factors, such as sufficient sleep, positive mental state and a light diet were largely found among the centenarian group. 

Findings from existing studies imply that the government should prepare for considerably increasing demand for long-term care in the near future [7,19]. However, the quality and availability of health resources vary greatly between coastal and inland areas as a result of their different levels of economic development. Even though social security provisions are implemented through the central government, how they are locally provided varies greatly. Geographical variations also exist in family support with a U-shaped relationship. Family support is weakest in mid-developed urban regions and highest in the most and least developed provinces. To better understand the health of the oldest-old cohorts in China, the aim of this paper is to provide a detailed analysis of the oldest-old in China, their health status and social determinants. Particular attention is given to the disparities and inequalities in health of, and health care for, the oldest-old age group between regions. 

## 2. Data and Methods

### 2.1. Data

As it is the first national-wide and longitudinal survey on the health of the older population, the Chinese Longitudinal Healthy Longevity Survey (CLHLS) provides the original data source for this study. Aiming to shed new light and a better understanding of the determinants of healthy longevity of human beings, the CLHLS has conducted seven waves of interviews (1998, 2000, 2002, 2005, 2008, 2011 and 2014) since its initiation in 1998, coving 23 of the 31 mainland provinces and representing approximately 85% of the total population [5]. To provide representative, meaningful and comparative information on populations at advanced age, the CLHLS uses internationally compatible questionnaires and has oversampled the oldest-old [23], which offers researchers a powerful database for studying the oldest-old group. The quality of the CLHLS data has been widely reported [24]. To reflect our research design and the effective sample size, data on the oldest-old from the 2011 CLHLS are used in this study. The total sample size of the oldest-old from the 2011 wave was 6395.

### 2.2. Indicators

Following extant research [1,25], in combination with the design of the CLHLS questionnaire, a set of indicators were chosen for data analysis (Table 1). Self-reported quality of life (QoL) and SRH are taken as the indicators of wellbeing in this study. The CLHLS required those interviewed to choose from very high, high, fair, low or very low to rate their QoL and health status. In line with previous research, Self-reported QoL is dichotomized into “high” (including very high, high and fair) and “low” (including low and very low). SRH is aggregated into “healthy” (including very good, good and fair) and “unhealthy” (including poor and very poor). SRH is controlled for in the study of QoL and QoL is controlled for in the study of SRH.

*Socioeconomic indicators*. In this study, information on demographic status, economic status, living conditions and care resources is used to reflect the socioeconomic status of the oldest-old groups under consideration. Specifically, gender, age, marital status and education level are selected to provide demographic characteristics. The oldest-old respondents’ former occupation, pension, and self-reported economic status are used to represent their economic status. The CLHLS questionnaire asked the respondents to rate their economic status as “very rich” “rich” “so-so” “poor” or “very poor”, compared with other local people. For ease of comparison, in this research, answers with “very rich” and “rich” are merged into “rich”, answers with “so-so” stay unchanged, and answers with “very poor” and “poor” are merged into “poor”. Housing tenure, having one’s own bedroom, the type of housing and living arrangements are selected to provide information on living conditions. Housing tenure is aggregated into “privately owned” combining purchased, self-built, inherited and welfare-oriented and “rented”. “Primary caregiver when being sick”, “primary medical insurance”, “whether elders get adequate medical services when they are sick” are used to reflect the care resources of the oldest-old.

*Physical condition.* Activities of daily living (ADLs) are used in this study as indicators of physical condition. The CLHLS questionnaire asked the respondents if they had any difficulty in bathing, dressing, continence, indoor transferring, using the toilet or feeding themselves. Answering with “yes” is valued as “1”, and answering with “no” is valued as “0”. From this, ADLs range from 0 to 6, with 0 indicating having no difficulty with daily living and 6 indicating having all the above difficulties with activities of daily living. 

*Health-related behaviors*. Smoking, drinking and exercising are selected as indicators of health-related behaviors.

*Indicators of “place”*. Residence and region are used to represent “place”. To reflect the unique “urban-rural” dual structure and the emergence of “new urban areas” in China in recent years, residence is categorized as city, town and rural areas in this study. Considering the notable regional differences in both economic development and social-cultural differences and in reference to the standard classification of the National Bureau of Statistics of China, the regions are classified as Eastern China, Central China, Western China and Northeastern China in this research. The effects of the interactions of residence and region are also considered to indicate further the influence of “place” on wellbeing.

### 2.3. Methods

In light of the over-sampling of older males, urban older residents and the oldest-old in the CLHLS, adjusted weights are applied in this study in any comparison between groups. Depending on the categorical or continuous attributes of variables, cross-tabulations or t-tests are used to conduct descriptive analysis on the relation of the independent variables with self-reported QoL and SRH. Binary logistic regression models are employed for both self-reported QoL and SRH respectively to estimate the role of geography and spatial variation in understanding the socioeconomic determinants of the health of the oldest-old. 

After eliminating samples with missing or incomplete information, 3807 cases were retained finally in the binary logistic regression analysis, with the number of excluded cases being 2588. Comparisons of the basic demographic information of the research samples before and after excluded cases were deleted are shown in Table 1. An analysis of the excluded cases detected a slightly bias towards males, town and rural districts and younger oldest-old groups. However, the differences between these two groups of samples are not obvious. Gender, residence and age are controlled in the following regression models. Therefore, deleting those samples with missing information does not impact the results of this study. A total of 6 models were ultimately used. Model 1 and model 4 evaluate the role of residence on self-reported QoL and SRH, while model 2 and model 5 estimate the effect of region on self-reported QoL and SRH respectively. Model 3 and model 6 further investigate the impact of the interaction of residence and region on self-reported QoL and SRH. 

IBM SPSS Statistics version 19 (IBM Inc, Armonk, NY, United States) was used to perform all the analysis and the Hosmer-Lemeshow (H-L) Test was used to evaluate the goodness of fit. The model has a good fit when the significance of the H-L Test is larger than 0.05 or the Accuracy is larger than 60% [26]. For ease of interpretation, all of the reference variables are in Italics in Table 2 and in brackets in Table 3 and Table 4. Odds ratios are used for analyzing results and in the discussion section. Only the significant results are shown in the tables and anyone who is interested in seeing the complete tables please contact the corresponding author.

## 3. Results

### 3.1. Descriptive Analysis

Descriptive statistics on the oldest-old with different levels of wellbeing are presented in Table 2. Gender is significantly correlated to SRH. Age, ADLs and occupation is significantly correlated to both self-reported QoL and SRH, and self-reported QoL and SRH are also closely linked. Significantly more males than females report their status as healthy. The respondents who report high QoL and being healthy are significantly older than the oldest-old who report low QoL and being unhealthy. The oldest-old respondents who report high QoL have a significantly higher percentage reporting being healthy compared to those who report low QoL, and the chances for the healthy oldest-old are also higher for reporting high QoL in comparison to the unhealthy oldest-old. The proportions having high QoL show a general downward trend with an increase in ADLs. Similarly, along with the increase in ADLs, a declining gradient is found with the proportion reporting being healthy. Significant differences exist in both QoL and health among the oldest-old with different previous occupations. Those whose employment had previously been in professional and technical occupations were more likely to report high QoL, while those whose previous employment had been in governmental, institutional or managerial occupations where more likely to report being healthy. By contrast, those whose previous occupations had been in the military were less likely to report high QoL and being healthy. 

Economic status, health-related behaviors, living conditions, and care and medical resources are all found to be significantly correlated to both self-reported QoL and SRH. In terms of economic variables, compared to the oldest-old respondents who have pensions, those without pensions are less likely to report having high QoL and being healthy. Both the proportions of high QoL and good health are lower with lower self-reported economic status compared to higher economic status. In terms of behaviors, the oldest-old who do not smoke or drink alcohol are less likely to report being healthy compared to those who smoke and drink, discussed further below. The oldest-old who do not undertake physical exercise are less likely to report having good health compared to those who do exercise. Housing tenure, conditions and living arrangements are also important. The oldest-old whose housing tenure is privately owned are more likely to report high QoL compared to those who are renters. Compared with the respondents who do not have their own bedroom, those who have them are more likely to report being healthy. The wellbeing of the oldest-old is also significantly different among those living in various types of housing. In the case of living arrangements, the oldest-old living with family are more likely to report high QoL than those living alone or living in a nursing house. The wellbeing of the oldest-old respondents also differs depending on their medical resources. Those having new cooperative medical insurance are less likely to report high QoL or good health compared to those having other types of medical insurance. Persons with adequate medical services are more likely to report high QoL and being healthy compared to those who do not have adequate medical services. As neither marital status nor levels of education were statistically significant indicators of self-reported QoL or SRH, they were excluded in the multivariate analyses in the following section.

Geographical factors also show variations. Respondents living in cities are more likely to report high QoL, followed by the oldest-old living in towns and rural areas. Although the p-value is marginally significant (0.089), the oldest-old living in cities are more likely to report good health than those living in rural areas. Those living in towns are least likely to report good health. There are significant differences in both QoL and health among the oldest-old from different regions. The oldest-old respondents living in Eastern China are more likely to report high QoL, followed by those living in Central China, Northeastern China, and Western China. Comparatively, the ratio reporting good health is highest among the oldest-old from Northeastern China, followed by the oldest-old from Eastern China, then Central China and finally Western China.

### 3.2. Binary Logistic Regression for Self-Reported QoL

The results of the binary logistic regressions for the odds of reporting high QoL are presented in Table 3. For ease of comparison, variables having significant impact in the QoL models are retained. SRH, economic status, care and medical resources, residence and the interaction of residence and regions are found to be significantly related to self-reported QoL. 

Model 1 shows that, among behavioral and socioeconomic factors, SRH, self-reported economic status and care resources are important in explaining the variation in self-reported QoL. Specifically, the reporting of high QoL when an oldest-old person reports being healthy is four times greater than when the oldest-old person reports being unhealthy. Compared to the respondents whose self-reported economic status is rich, people with medium economic status are much less likely to report high QoL. The odds show the oldest–old respondents with poor economic status are least like to report high QoL. In comparison to the oldest-old whose primary caregivers are family members when they are sick, respondents who dependent on others such as neighbors, friends or social service providers as their primary caregivers, the oldest-old who have live-in caregivers as primary caregivers, and the oldest-old who have nobody to take care of them are less likely to report high QoL. Compared to respondents who have adequate medical services, those whose medical services are inadequate are much less likely to report high QoL. The effect of residence on self-reported QoL is significant. Both the oldest-old living in towns and rural areas are significantly less likely to report high QoL, and the oldest-old from rural areas have the lowest odds ratio.

Model 2 investigates the effects of region on self-reported QoL with a list of other factors being controlled. Similar to the findings in model 1, SRH, self-reported economic status and care resources are significant in explaining self-reported QoL. The oldest-old persons from different regions in China failed to show any significant difference in their QoL when other factors are controlled.

Model 3 further explores the impact of “place” on QoL by adding the interaction items of residence and region. The effects of other factors stay similar to model 1 and model 2. With respect to the effects of the interaction items, compared to the oldest-old respondents living in Eastern cities, the oldest-old living in Western rural, Central rural and Central towns are all significantly less likely to report having high QoL.

### 3.3. Binary Logistic Regression for SRH

The results of the binary logistic regression models for odds ratios of reporting good health are presented in Table 4. Variables having significant impact in the health models are retained.

Gender, age, self-reported QoL, ADLs, economic status, health related behaviors including drinking and exercising, medical resources, region and the interactions of residence and region are all significantly related to SRH. Model 4 shows that, although the impact of residence on SRH is not significant in contrast to the self-reported QoL models where it was significant, demographic status, physical conditions, economic status, health-related behaviors and care resources are important in explaining health status. Specifically, males are more likely to report being healthy than females. For the oldest-old, the chances of reporting good health are significantly higher with an increase in age, which is similar to former studies [8,23]. The oldest-old with high QoL are four times more likely to report good health, compared to those who reported low QoL. Compared to the oldest-old who have no difficulties with daily living, there is a declining gradient in the probability of reporting good health with the increase in the values of ADLs, except for the insignificant difference for the oldest-old respondents with one limitation in ADLs.

Similarly, there is also a declining gradient in the possibility of having good health with the decrease in self-reported economic status. Contrary to our initial expectations, the oldest-old who do not drink alcohol are less likely to report good health compared to those who do drink alcohol. We speculate that there are two possible reasons for this outcome. For one thing, the oldest-old with serious health conditions might have quit unhealthy behaviors, leaving only those who report their health as good who are still drinking. A second possibility is that the CLHLS questionnaire only asks if a person drinks, but does not ask about how much they drink, how often they drink or if they drank in the past. In comparison with the respondents who do exercise, those who do not do exercise are significantly less likely to report good health, which could possibly be because those in poor health cannot undertake exercise. Compared to the oldest-old who have adequate medical insurance, those who do not have adequate medical insurance are significantly less likely to report good health.

Model 5 estimates the effect of region on the chance of reporting good health. The significance and values of the odds ratios are similar for the majority of variables compared to model 1. The only exception is that the odds ratio of the oldest-old with one limitation with ADLs is marginally significant, resulting in a declining gradient of odds ratios of reporting good health with an increase in the number of ADLs. Compared to the oldest-old living in Eastern China, those living in Western China are less likely to report good health. The respondents from Northeastern China are 0.8 times more likely than those living in Eastern China to report good health.

Model 6 further investigates the impact of the interactions of residence and region on SRH. The effects of other variables remain similar to those in models 4 and 5. Including the interaction terms, the oldest-old from Northeastern cities are 1.8 times more likely to report good health than those from Eastern cities. By contrast, in comparison to the oldest-old living in Eastern cities, those living in Central towns are less likely to report good health, which is also marginally significant. 

## 4. Discussion

This study investigates the distributions of both QoL and health status of the oldest-old in various regions of China, and its possible socioeconomic determinants, with a particular attention to the comparison across different types of residence and regions. The results from the analysis of CLHLS data provide interesting and novel insights. It seems that significant differences in both QoL and health exist among the oldest-old with different occupations, pension programs, economic status, housing conditions, care resources and social service. Economic status, social services, and indicators of “place” are all significant determinants of both QoL and health. The chances of reporting high QoL and health decrease with the descending of income level. Compared to those who do have adequate medical services, the oldest-old who do not are significantly less likely to report high QoL and being healthy. Although being insignificant in the models of SRH, the oldest-old having neighbors, friends or social service as primary caregivers when they are sick, those having nobody to take care of them, or those having live-in caregivers all have a much lower probability of reporting high QoL. The oldest-old living in towns and rural areas are less likely to report high QoL compared to those living in cities. The interactions between residence and regions further indicate that the oldest-old living in Central rural, Western rural or Western town areas are less likely to report high QoL compared with those living in Eastern cities. The significant impacts of “place” on SRH lie in that, compared to the oldest-old in Eastern China, the oldest-old in Western and Northeastern China are respectively less and more likely to report good health. Specifically, the interaction items further indicate that the oldest-old in Western town areas are significantly less likely to report good health, and the oldest-old in Northeastern city areas are significantly more likely to report good health, compared to their counterparts living in Eastern cities.

To date, studies focusing on the social determinants of health and wellbeing among the oldest-old in China are quite limited. However, as the oldest-old cohorts are just at the beginning of their projected considerable growth, and China will most likely become the most prominent country in terms of growth of the oldest-old cohorts [5], such research must become ever more important. The role of geography in understanding the distribution and social determinants of the oldest-old in China remains largely unexplored, although in light of the significant geographical variations in health resources, existing studies imply the urgent need of more targeted policy to satisfy the increasing demand for elderly care in the near future [27]. The originality of this current research is that it is one of the first few studies concentrating on the social determinants of life for the oldest-old group to pay special attention to the role of geography. Being one of very few studies to analyze the oldest-old population on a national scale is another merit of this paper. The following discussion summarizes the principal foci of the study and offers some potential explanations for the main findings.

### 4.1. Factors Explaining the QoL and SRH of the Oldest-Old are in General the Same Factors that Explain QoL and Health Status of the Older Population.

The findings of this study suggest that the socioeconomic determinants of the QoL and health status of the oldest-old are similar to those for the older population. The chances of reporting high QoL and good health for the oldest-old reduce with descending income level, which is similar to Feng et al.’s findings [13]. Lacking sufficient medical service will significantly reduce the probability of reporting high QoL and of being healthy, also reported in other studies on the older population [22,28]. Although applying different classifications from that in this study, its findings of significant differences among cities, towns and rural areas in chances to report high QoL reflect Zeng et al.’s research which also mentions the significant impact of residence on older population’s QoL [23]. Moreover, the significant impacts of age, gender and health behaviors on older people’s health are also reported elsewhere [29]. In this study, the finding that the oldest-old who do not drink are significantly less likely to report good health than the oldest-old who do drink is rather unexpected. However, the literature indicates that the impacting factors of health are synthetic and diverse [27,30]. Amounts and duration of drinking are not reported, and there might be a chance that the oldest-old who do drink and live over 80 years are those who have other factors such as genetics, economic status and the like that favor good health.

### 4.2. Potentially the Interaction Effect of Residence and Region Seems to Matter More Than Each of the Individual Factors

The interaction effect of residence and region further indicates the impact of geography on both QoL and health in a more detailed way. Partly because of the unique urban-rural dual structure in both socioeconomic development and administration [5], the oldest-old who live in rural areas were significantly less likely to report high QoL compared to the oldest-old living in cities. As an administrative unit between city and country [31], the town has a medium effect on QoL, with the oldest-old living in towns being less likely to report high QoL compared to the oldest-old living in cities, whilst being more likely to report high QoL compared to the oldest-old living in rural areas. Interestingly, although the effect of region on QoL is insignificant, the interaction of residence and region suggests that the city-town-rural difference in chances to report high QoL may lie in the fact that the oldest-old living in Central rural areas, Western towns, and Western rural areas are much less likely to report high QoL compared to the oldest-old in Eastern cities. According to the report of the National People’s Congress of the People’s Republic of China, economic development, social welfare, infrastructure and physical environment in Western China have all been improved substantially ever since the implementation of China’s Western Development Strategy in 2000. However, to promote further the construction of small cities and villages is the focus of the next few years [32]. The *Sannong Issues* (i.e., issues concerning the development of agriculture, farmers and rural areas) have been the primary focus of China’s central government ever since the Rise of Central China Strategy was initially proposed by the former prime minister Wen Jiabao in 2004 [33]. The remained gap in socio-economic development in Western towns, Western rural areas and Central rural areas may offer some reasonable explanations to the lower chances of the oldest-old from these areas in reporting good QoL, compared to those from the Eastern cities. Comprehensively, the interactions of region and residence seem to matter more by providing us with extra information in understanding the impact of geography on QoL, although the Accuracy rate of model 3 equals to that of model 1. 

By contrast, in the case of SRH, there are significant differences in the chances of reporting good health across regions. The oldest-old respondents living in Western China were much less likely to report good health and the oldest-old living in Northeastern China were more likely to report good health, compared to those from Eastern China. The lower level of both life expectancy at birth and disability-free life expectancy in Western China and the higher level of these items in Northeastern China [34,35], compared to that of Eastern China, might provide some clues for the higher possibility for the oldest-old in Northeastern China in reporting good health and the lower possibility of the oldest-old in Western China in reporting good health. The percentage of the oldest-old is higher in Northeastern China and lower in Western China compared to Eastern China [35]. Although the differences in chances of reporting good health for city, town and rural areas are insignificant, the interactions of residence and region further suggest that the differences across regions may be because, compared to Eastern cities, the oldest-old living in Western towns are significantly less likely to report good health, and the oldest-old living in Northeastern cities are significantly more likely to report good health. The lower percentage of the oldest-old in Western towns (12%) in 2011 compared to that of Western rural areas (15.5%), and the higher percentage of the oldest-old in Northeastern cities (15.6%) compared to that of Northeastern towns (13.6%) and Northeastern rural areas (14.1%), and the higher chance of reporting good health with the increase of age for the oldest-old, may provide some reasonable explanations [4]. Moreover, the Accuracy rate in model 6 is 83.6 %, which is bigger than 83.3 % in model 4 and 83.5 % in model 5. This may potentially indicate that interaction of residence and region matter more for health than each of the individual factors.

### 4.3. From a Policy Perspective, the Significant Urban-Town-Rural and Regional Differences Suggest a more Geographical Targeted Approach to the QoL and Health Status of the Oldest-Old is likely to be Required.

To improve the QoL and health status of the oldest-old cohorts, the Ministry of Civil Affairs of the People’s Republic of China called on governments to adopt old-age allowance policies as early as 2009, and the Law of the People’s Republic of China on Protection of the Rights and Interests of the Elderly also encourages local government to establish old-age allowances aimed at the oldest-old [36]. However, the coverage rate of older persons having access to old-age allowance was still low. Only 39.69% of the oldest-old had access to old-age allowance in 2014 [37]. The amount of old-age allowance was still at a low level, ranging from 30 RMB to 800 RMB per person per month for different regions, and being 300 RMB per person per month for the majority of provinces in 2014 [38]. Moreover, the orientations of old-age allowance in most provinces are unclear, resulting to very limited profit range and very little effect [39]. The logistic analyses in this study suggests that to improve the economic status of the oldest-old and to make sure that they acquire sufficient medical care during sick are important in improving both the level of self-reported QoL and SRH. Moreover, this study further indicate that the oldest-old having family caregivers are more likely to report high QoL than the oldest-old having other caregivers, suggesting the primary role of family in elderly care in current China. In light of the sharp decrease in the availability of adult children being caregivers [5], this study calls for more supportive policies in promoting family care to improve the QoL of the oldest-old. This current study suggests that in explaining the distribution of QoL and health, even after controlling for individual demographic factors, other than socioeconomic variables, the impact of residence, region and the interactions between them are noticeable. Therefore, a more geographical targeted approach in policymaking should be employed to promote QoL and good health in a more effective way. Policy making can take place at various scales including the community, municipal, provincial and national levels. For example, at the national level, both the pension and health care insurance systems should be unified in the future for older people living in rural and urban areas. At the provincial and municipal levels, the variance of social welfare benefits should take into account factors such as living and health care expenses of older populations in various regions. Evaluation of care needs and service provision can be implemented at the community level.

### 4.4. Limitations 

Several limitations of this study should be mentioned. First, limited by the scale of the oldest-old samples across the years, the study only used the 2011 cross-sectional data. Therefore, we should not draw any causal or longitudinal inferences from the findings of this study. The results of our regression model only indicate the extent of correlations. Second, also limited by the scale of the samples, only residence, regions and their interactions are considered to explore the role of geography in understanding the distribution of both QoL and health. In view of the significant differences in China in terms of economic development, social structure, and physical environment across different scale of units such as provinces, cities, and neighbourhoods, it is important to conduct analysis across a wide range of administrative units in the future. Third, because many of the oldest-old are low-income widowed women [34], who are likely to have high demands for financial support and care resources, the health status and determinant of this vulnerable group should be focused in our follow-up research.

## 5. Conclusions

This study investigated the socioeconomic determinants of the QoL and health status of the oldest-old, with special attention paid to the role of residence and region in explaining the distributional aspects of QoL and health. The results indicate that, after controlling for individual demographic and health behavior variables, both economic level and social welfare have a significant effect on QoL and health. There are also significant differences in QoL between city, town and rural areas, with the oldest-old respondents living in Central rural, Western town and Western rural areas being significantly less likely to report high QoL compared to the oldest-old living in Eastern city areas. Similarly, significant differences in health exist among Eastern China, Western China and Northeastern China, with the oldest-old from Western towns being significantly less likely to report good health, and the oldest-old from Northeastern cities being significantly more likely to report good health, in comparison to the oldest-old from Eastern cities. The results of this study indicate that socioeconomic factors that explain QoL and health status of the older population are in general factors that explain the QoL and health status of the oldest-old cohorts. Comparison between models suggests that the interaction effect of residence and region seem to matter more than each of the individual factors, in providing us with more detailed information on the role of geography in explaining QoL and the health of the oldest-old. At a time when the oldest-old cohorts in China are at the beginning of their projected growth, the findings of this study are vital evidence for policy makers, providing them with more information on the urgency of making more geographically targeted policy to improve more effectively the QoL and health of the oldest-old population.

## Figures and Tables

**Table 1 ijerph-16-00601-t001:** Comparisons of samples before and after excluded cases were deleted.

	Total Samples (No. of Respondents/Percentage)	Samples with Complete Information (No. of Respondents/Percentage)
**Gender (%)**		
Female	3651 (57.1)	2128 (55.9)
Male	2744 (42.9)	1679 (44.1)
**Residence (%)**		
City	921 (14.4)	472 (12.4)
Town	1746 (27.3)	1085 (28.5)
Rural	3728 (58.3)	2250 (59.1)
**Age (years)**		
Max	106	106
Min	80	80
Mean	91.4	90.7

Note: Although a slight bias existed towards males, town and rural districts and younger oldest-old groups, the differences between the two groups of samples are not obvious. Gender, residence and age are controlled in the regression models, deleting those samples with missing information does not impact the results of this study.

**Table 2 ijerph-16-00601-t002:** Descriptive information on the self-reported QoL and SRH of the oldest-old.

	Self-Reported QoL			SRH		
	Bad	Good	*p* Value	Unhealthy	Healthy	*p* Value
**Gender**						
*Female*	5.50	94.50	0.559	20.00	80.00	**0.004**
Male	5.00	95.00		16.40	83.60	
**Age** (80–103)	89.71(6.39)	90.74(6.94)	**0.04**	89.71(6.65)	90.9(6.96)	**<0.001**
**SQOL**						
*Bad*				61.50	38.50	**<0.001**
Good				16.00	84.00	
**SRH**						
*Unhealthy*	17.50	82.50	**<0.001**			
Healthy	2.50	97.50				
**Activities of Daily Living (ADLs)**						
*0*	4.90	95.10	**<0.001**	15.00	85.00	**<0.001**
1	3.30	96.70		15.00	85.00	
2	6.70	93.30		24.20	75.80	
3	3.30	96.70		25.30	74.70	
4	5.60	94.40		33.60	66.40	
5	11.90	88.10		49.40	50.60	
6	15.90	84.10		55.10	44.90	
**Pension**						
*Yes*	3.90	96.10	**0.01**	15.80	84.20	**0.004**
No	5.90	94.10		19.70	80.30	
**Self-reported economic status compared with other local people**						
*Rich*	0.60	99.40	**<0.001**	8.80	91.20	**<0.001**
So-so	2.40	97.60		16.90	83.10	
Poor	23.60	76.40		37.10	62.90	
**Drinking**						
*Yes*	3.80	96.20	0.082	11.50	88.50	**<0.001**
No	5.50	94.50		19.80	80.20	
**Exercising**						
*Yes*	4.30	95.70	0.065	11.90	88.10	**<0.001**
No	5.70	94.30		21.70	78.30	
**Housing tenure**						
*Privately owned*	5.00	95.00	**0.043**	18.20	81.80	**0.247**
Rented	8.00	92.00		21.10	78.90	
**Who takes care of you when you are sick**						
*Family*	4.70	95.30	**<0.001**	18.20	81.80	0.113
Other relatives	6.00	94.00		17.10	82.90	
Neighbors, friends or social service	21.90	78.10		18.80	81.30	
Live-in caregiver	6.70	93.30		23.30	76.70	
Nobody	28.30	71.70		32.60	67.40	
**Medical insurance**						
*Urban residents’ basic medical insurance*	2.30	97.70	**0.001**	13.70	86.30	**0.010**
Urban employee basic medical insurance	2.00	98.00		12.80	87.20	
New rural cooperative medical insurance	6.10	93.90		19.50	80.50	
Gongfei medical insurance	2.40	97.60		18.20	81.80	
Commercial medical insurance		100.00		25.00	75.00	
**Do you get adequate medical service when sick?**						
*Yes*	4.20	95.80	**<0.001**	17.10	82.90	**<0.001**
No	23.20	76.80		40.50	59.50	
**Residence**						
*City*	2.20	97.80	**0.001**	15.20	84.80	0.089
Town	5.40	94.60		19.30	80.70	
Rural	6.10	93.90		18.80	81.20	
**Region**						
*Eastern China*	4.20	95.80	**0.003**	17.10	82.90	**0.001**
Central China	4.90	95.10		17.80	82.20	
Western China	7.40	92.60		22.20	77.80	
Northeastern China	5.60	94.40		11.80	88.20	

Note: Marital status, education, occupation, smoking, inquiries on if the elders had their own bedroom, house category, living arrangements are not shown in this table for being insignificant in the descriptive analysis or in the following regression models.

**Table 3 ijerph-16-00601-t003:** Binary logistic regression for odds ratios of self-reported QoL (Bad QoL as reference).

	Model 1 (CI)	Model 2 (CI)	Model 3 (CI)
**Healthy (Unhealthy)**	5.02(3.57–7.08) ***	5.14(3.64–7.25) ***	5.09(3.60–7.20) ***
**Self-reported economic status compared with other local people (Rich)**			
So-so	0.28(0.10–0.78) *	0.27(0.10–0.76) *	0.28(0.10–0.78) *
Poor	0.03(0.01–0.10) ***	0.03(0.01–0.09) ***	0.04(0.01–0.10) ***
**Who take care of you when you are sick (Family)**			
Neighbors, friends or social service	0.38(0.14–1.07) #	0.36(0.13–1.00) #	0.38(0.13–1.06) #
Live-in caregiver	0.33(0.10–1.10) #		0.32(0.10–1.02) #
Nobody	0.35(0.15–0.81) *	0.34(0.15–0.78) *	0.33(0.14–0.75) **
**Don’t have adequate medical service (Have)**	0.58(0.37–0.89) *	0.57(0.37–0.88) *	0.58(0.38–0.90) *
**Residence (City)**			
Town	0.45(0.23–0.89) *		
Rural	0.38(0.19–0.73) **		
**Region*Residence (Eastern City)**			
Central Rural			0.46(0.18–1.15) #
Western Town			0.48(0.20–1.15) #
Western Rural			0.45(0.18–1.13) #
Sig. of H-L Test	0.86	0.26	0.66
Accuracy (%)	94.70	94.50	94.70

**Note:** *** *p* < 0.001; ** *p* < 0.01; * *p* < 0.05; # *p* < 0.1. Anyone interested in seeing the complete tables is welcome to contact the corresponding author.

**Table 4 ijerph-16-00601-t004:** Binary logistic regression for odds ratios of self-reported health (Unhealthy as reference).

	Model 4 (CI)	Model 5 (CI)	Model 6 (CI)
**Male (Female)**	1.18(0.98–1.43) #	1.17(0.97–1.42) #	1.18(0.97–1.42) #
**Age**	1.06(1.04–1.07) ***	1.06(1.04–1.07) ***	1.06(1.04–1.07) ***
**High quality of life (Low)**	5.06 (3.59–7.14) ***	5.04(3.57–7.11) ***	5.00(3.54–7.07) ***
**ADLs (0)**			
1		0.76(0.57–1.02) #	0.76(0.57–1.01) #
2	0.50(0.33–0.75) ***	0.48(0.32–0.72) ***	0.48(0.32–0.72) ***
3	0.42(0.25–0.71) ***	0.39(0.23–0.65) ***	0.38(0.23–0.65) ***
4	0.30(0.20–0.46) ***	0.28(0.18–0.43) ***	0.29(0.19–0.44) ***
5	0.18(0.12–0.25) ***	0.17(0.12–0.25) ***	0.17(0.12–0.24) ***
6	0.16(0.09–0.27) ***	0.14(0.08–0.24) ***	0.14(0.08–0.25) ***
**Self-reported economic status compared with other local people (Rich)**			
So-so	0.56(0.42–0.75) ***	0.57(0.43–0.77) ***	0.58(0.43–0.77) ***
Poor	0.32(0.23–0.46) ***	0.33(0.23–0.47) ***	0.34(0.24–0.48) ***
**Not drink (Drink)**	0.65(0.49–0.86) **	0.65(0.49–0.86) **	0.65(0.49–0.87) **
**Not exercise (Exercise)**	0.67(0.54–0.84) ***	0.65(0.52–0.81) ***	0.65(0.52–0.81) ***
**Don’t have adequate medical service (Have)**	0.64(0.45–0.89) **	0.65(0.46–0.91) *	0.65(0.46–0.92) *
**Region (Eastern China)**			
Western China		0.69(0.56–0.86) ***	
Northeastern China		1.88(1.11–3.19) *	
**Region*Residence (Eastern City)**			
Western Town			0.67(0.43–1.04) #
Northeastern City			2.80(1.18–6.63) *
Sig. of H-L Test	0.79	0.46	0.99
Accuracy(%)	83.30	83.50	83.60

Note: *** *p* < 0.001; ** *p* < 0.01; * *p* < 0.05; # *p* < 0.1. Anyone interested in seeing the complete tables is welcome to contact the corresponding author.

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
