# Peer review of "Understanding the Wellbeing of the Oldest-Old in China: A Study of Socio-Economic and Geographical Variations Based on CLHLS Data"

_ijerph, 2019, doi:10.3390/ijerph16040601_

Round 1
Reviewer 1 Report
This study focused on analysing the well-being among oldest-old in China. This paper was quite interesting, and this can help different professionals to promote well-being among elder people. However, few issues should be further concerned.
Some tables were too long, and they should be further revised in order to demonstrate a clear presentation.
Referring to Point 1, it is suggested to provide a summary of findings in appropriate places of the part of "Results"
In the part of "Discussion", the authors should provide more explanations on the current results.
Also, the results of interaction effects of residence and region were also quite important, and the authors are suggested to further elaborate them.
Based on the current findings, I think the authors can provide more discussion regarding the implications of this results. For example, how do the officials generate appropriate policies to promote oldest-olds' well-being based on the results?
I hope the above observation is useful, and thanks for giving me this opportunity to review this manuscript.
Author Response
We thank the reviewer for these valuable comments on the manuscript. As authors, we have made revisions to the paper based on these suggestions. The revisions cover the following aspects:
1. Some tables were too long, and they should be further revised in order to demonstrate a clear presentation.
We have recreated Table 3 and 4, and only show the statistically significant results in the tables. We have added “only the significant results are shown in the tables and anyone who is interested in seeing the complete tables please contact the corresponding author”, in Line 237-239.
2. Referring to Point 1, it is suggested to provide a summary of findings in appropriate places of the part of "Results"
We provide a summary of findings in the results section. Please see line 246-248, 262-263, 303-304, 336-338 for details.
3. In the part of "Discussion", the authors should provide more explanations on the current results. Also, the results of interaction effects of residence and region were also quite important, and the authors are suggested to further elaborate them.
We have added more explanations on the results in the discussion section, with special focus on interpreting the results on the interaction effects of residence and region. Please see line 444-454, 461-468, 472-477 for more details.
4. Based on the current findings, I think the authors can provide more discussion regarding the implications of this results. For example, how do the officials generate appropriate policies to promote oldest-olds' well-being based on the results?
Line 489-499, 503-509. We have added policy implications for this study “Policy making can take place at various scales including the community, municipal, provincial and national levels. For example, at the national level, both the pension and health care insurance systems should be unified in the future for older people living in rural and urban areas. At the provincial and municipal levels, the variance of social welfare benefits should take into account factors such as living and health care expenses of older populations in various regions. Evaluation of care needs and service provision can be implemented at the community level.”
Reviewer 2 Report
This is an interesting work about quality of life in China
I found it worth to be published however I would like to point some things that should be improved.
The introduction is too long, it should be reduced
Tables 2,3,4 have a lot of information and it is difficult to follow. May be the information could be grouped in smaller tables or a combination of Tables and Figures.
Author Response
Many thanks to the reviewer for these valuable comments on the manuscript. The authors have made revisions to the paper based on the suggestions covering the following aspects:
1. The introduction is too long, it should be reduced
Line75-77, 91-93, 118-122. We have cut down the introduction section.
2. Tables 2,3,4 have a lot of information and it is difficult to follow. May be the information could be grouped in smaller tables or a combination of Tables and Figures.
We have recreated the tables and only show the statistically significant results in Table 3 and 4. We have added “only the significant results are shown in the tables and anyone who is interested in seeing the complete tables please contact the corresponding author” in Line 237-239.
Round 2
Reviewer 1 Report
No further comment, and it is suggested to check potential typos before further process.
Reviewer 2 Report
I fins that the authors have answered the questions and improved the manuscript. Therefore it can be published in the present form